# Responses of Dairy Buffalo to Heat Stress Conditions and Mitigation Strategies: A Review

**DOI:** 10.3390/ani13071260

**Published:** 2023-04-05

**Authors:** Francesca Petrocchi Jasinski, Chiara Evangelista, Loredana Basiricò, Umberto Bernabucci

**Affiliations:** 1Department of Agriculture and Forests Sciences, University of Tuscia-Viterbo, via San Camillo De Lellis, snc, 01100 Viterbo, Italy; 2Department for Innovation in Biological Agro-Food and Forest Systems, University of Tuscia-Viterbo, via San Camillo De Lellis, snc, 01100 Viterbo, Italy

**Keywords:** dairy buffalo, heat stress, feeding, milk yield and composition, reproduction, mitigation strategies

## Abstract

**Simple Summary:**

Buffalos are mainly reared in Asia, but little by little, they are becoming important in other part of the world such as Africa and Europe. Due to increases in temperature and climate change, most dairy animals are affected by heat stress (HS), a condition in which they cannot dissipate an adequate quantity of endogenous heat. Heat stress negatively affects clinical parameters, reproductive performance, and milk yield and its characteristics, with a huge economic impact on farmers and the buffalo milk industry. Nevertheless, there are some strategies to mitigate the effect of HS by adding antioxidant substances to their diet, modifying the content of fiber and protein of the diet, or providing cooling systems to the buffalo such as shade, fans, sprinklers, and pools.

**Abstract:**

Increases in temperature and the greater incidence of extreme events are the consequences of the climate change that is taking place on planet Earth. High temperatures create severe discomfort to animal farms as they are unable to efficiently dissipate their body heat, and for this, they implement mechanisms to reduce the production of endogenous heat (reducing feed intake and production). In tropical and subtropical countries, where buffalo breeding is more widespread, there are strong negative consequences of heat stress (HS) on the production and quality of milk, reproduction, and health. The increase in ambient temperature is also affecting temperate countries in which buffalo farms are starting to highlight problems due to HS. To counteract HS, it is possible to improve buffalo thermotolerance by using a genetic approach, but even if it is essential, it is a long process. Two other mitigation approaches are nutritional strategies, such as the use of vitamins, minerals, and antioxidants and cooling strategies such as shade, fans, sprinklers, and pools. Among the cooling systems that have been evaluated, wallowing or a combination of fans and sprinklers, when wallowing is not available, are good strategies, even if wallowing was the best because it improved the production and reproduction performance and the level of general well-being of the animals.

## 1. Buffalo Introduction

*Bovidae* is a family rank that contains nine genera, among which there is the genus *Bubalus*. This genus, in turn, includes five species: *B. arnee* also known as wild water buffalo which lives in Asia; *B. depressicornis* and *B. quarlesi* known as lowland anoa and mountain anoa, respectively, which live only in Indonesia; *B. mindorensis* known as Tamarraw or Mindoro which lives only on Mindoro Island in the Philippines; and *B. bubalis* known as water buffalo [1]. This last species includes two different subspecies: *B. bubalis bubalis* known as river buffalo and *B. bubalis kerebau* known as swamp buffalo. The first one is bigger than the last one and is raised especially for milk production. A river buffalo’s body weight is between 450 and 1000 kg compared to 320–450 kg of swamp buffalo which is raised mainly for work purposes. In addition, these two subspecies are interfertile with their progeny containing 49 chromosomes given that the river buffalo is 2*n* = 50 chromosomes and the swamp buffalo is 2*n* = 48 chromosomes [2]. Globally, there are 123 breeds of water buffalo [3], but only three breeds are widely distributed worldwide: Murrah, Nili-Ravi, and Mediterranean Italian thanks to their high milk yield [1]. FAO [4] reported that the population of buffalo at the worldwide level amounted to 203,939,158 heads in 2021.

The introduction of this animals in Italy is not yet well known, but year by year, it has begun to become more important to the economy of the State, so much that in 2000, the “Mediterranean Italian” buffalo breed was officially recognized by the Ministry of Agriculture [5]. This breed has a thick skin, is black to reddish or slate gray in color, and becomes lighter in the belly area. The horns are brown, symmetrical, 50–60 cm long, and directed sideways and backwards. The average weight of the female is around 650 kg despite the weight of the male being between 700–800 kg [6]. Mediterranean Italian buffalo breed is raised almost exclusively for milk production, which is mainly used to produce mozzarella cheese, and in central-south Italy, milk is used to produce a specific mozzarella cheese called “Mozzarella di Bufala Campana PDO”. Productivity is about 1800–3500 kg per lactation [7] and milk production at a global level is from 600 to 4500 kg per lactation [8]. FAO [4] reports worldwide milk production amount to have been 137,761,642.79 L in 2021, which corresponds to 15% of the total milk yield. This result puts buffalo milk in second place to milk’s ranking around the world, it is preceded only by bovine milk. The nutritional quality of buffalo milk is better than cow’s milk; in fact, buffalo milk contains about 8.2% fat and 4.3% protein with respect to the 3.8%fat and 3.4%protein of cow’s milk. Moreover, buffalo milk also has a higher content of minerals compared to cow’s milk [9].

Today, Mediterranean Italian buffalo management is mainly intensive. Dairy buffalos are kept loose in paddocks close to the milking room, where the animals are submitted to udder control and mechanically milked twice a day. Some females are subjected to estrus induction to obtain calving before spring (about 50% fertility) as the milk pays out more in spring and summer according to consumer demand. In intensive systems, dairy buffalos normally receive a total mixed ration (TMR) composed of maize silage, concentrates, hay, straw, and sometimes by-products [10].

In Asia, the main production systems are extensive, where the animals live in pastures and grazing provides 50–60% of the feed resources, or semi-intensive, where the feedstuff used is wheat straw, chopped green roughages, concentrates, cotton seed cakes, and rice by-products, and they are given in the open air along feeding ways [10,11].

The production of milk and its chemical composition are affected by several factors such as breed, genetics, the number of lactations, the days of lactation, the lactation phase, feeding, season, health status, etc. [12,13].

The objective of this review is to analyze how the season and, in particular, the hot season (namely heat stress, HS) interfere with buffalo livestock and especially the effect that HS has on the dairy buffalo performances.

## 2. Heat Stress Introduction

The demand for livestock products is growing rapidly, driven by population and income growth plus urbanization. Simultaneously, livestock production is facing increasing pressure from climate change effects, such as increasing temperatures, more variable precipitation patterns, more frequent extreme events, and increasing carbon dioxide concentration [14]. The global temperature has increased by 1.5 °C above pre-industrial levels [15]. Moreover, the increasing temperature is accompanied by water scarcity, and this condition leads to a more severe situation of alert [16]. Given the current climate scenario and not very encouraging future estimates [17,18], we must expect farming to be increasingly adversely affected by the effects of HS. Heat stress can be simply defined as a condition that occurs when an animal cannot dissipate an adequate quantity of endogenous heat, whether it is produced or absorbed by the body, to maintain body thermal balance [19]. This can happen when the ambient temperature and humidity are high and can cause a range of negative effects on the animal’s health and productivity. Thermal stress is triggered when environmental conditions exceed the upper or lower critical temperature of domestic animals requiring an increase in basal metabolism to deal with the stress [20]. Heat stress occurs when the ambient temperature is above the thermoneutral zone.

It is well known that HS can cause a range of negative effects on dairy cattle, including decreased milk production, reduced feed intake, reduced reproductive performance, and increased susceptibility to diseases [21]. In addition, HS can cause physiological changes such as an increased heart and respiration rate, increased blood flow to the skin, and increased sweating which can lead to a loss of electrolytes, dehydration, and inflammation. For these reasons, HS has a huge economic impact on the global dairy industry.

Little information is available on the effect of HS in buffalo reared under intensive conditions. Buffalos are widespread in many areas of the world, mainly in Asia, some Mediterranean countries, and some Eastern European and Latin American countries [22]. Therefore, buffalos are adapted to different climates. In particular, due to their morphological, anatomical, and behavioral characteristics, buffalos are better adapted to hot and humid climates [22]. Compared to cows, it appears that buffalos are more tolerant to HS and can better handle tropical climates, with fewer negative effects on production and physiology. However, they exhibit signs of greater distress when exposed to direct solar radiation [22,23]. The body temperature of buffalo is slightly lower than that of cattle, but buffalo skin is usually black, so their body absorbs a large amount of solar radiation due to their dark skin and sparse hair, and, in addition, buffalo have a less efficient evaporative cooling system due to their rather poor sweating capacity compared with cattle [22]. In fact, buffalo skin has one-sixth of the density of sweat glands compared to cows; thus, buffalo dissipate heat poorly by sweating [22]. The maximum number of sweat glands/mm^2^ on buffalo skin was observed the in head dorsal followed by the abdomen dorsal, tail dorsal, neck dorsal, and thorax dorsal areas [24]. The distribution of sweat glands, the dark color of the skin, and the sparse body hair adversely affect heat tolerance in these animals.

The greater presence of buffalo is mostly found at latitudes characterized by a hot–humid climate (tropical and subtropical areas) and for this reason, they have developed defensive mechanisms over time such as a large quantity of melanin and sebaceous glands which, respectively, prevent ultraviolet rays from being able to pass through the dermis [25] and, through the sebum secreted and dissolved by high temperatures, reflect the sun’s rays [26]. These glands secrete sebum, a fatty substance that covers the surface of the skin and coats it with a lubricant, making it slippery to water and mud. This fatty sebum, together with the thick upper horny layer of the skin, prevents the water and solutes it contains from being absorbed by the skin. In this way, the animal is protected from the harmful effects of any deleterious chemical compounds present in the water. Furthermore, the sebum layer melts during the hot season and becomes more shiny reflecting many of the Sun’s rays, thus relieving the animal from excessive external thermal load.

Buffalos are more heat-stressed (HSed) when they are prevented from displaying their adaptive behavioral traits such as seeking shelter, wallowing, and/or diving into water [27]. It has been shown that buffalos are better able to regulate their body temperature if left to splash around in pools of water rather than using showers as a cooling system [28]. It is not always possible to have pools; however, like any other animal, they still need to have appropriate measures for HS management to maintain their productivity and health.

The exposure of buffalo to hot conditions causes a series of changes in their biological functions which includes a decrease in feed intake, the efficiency and utilization of their diet, disturbances in the metabolism of water, proteins, energy and mineral balances, enzymatic reactions, hormonal secretions, and blood metabolites [29], a reduction in milk yield and quality [30], and a reduction in the manifestation of heat and therefore fertility [31]. The effects of HS on the quality of production are less evident than in cattle [30]; the negative impacts of HS on various biological functions are discussed in more detail later in the paper.

High air temperature and humidity are the main factors responsible for variations in the physiological reactions of animals, and these variations can be different depending on the species and breed [21,32]. As reported in the literature [33], thermal parameters are the key factors used to calculate heat transfer: these include temperature, humidity, wind speed, and solar radiation. The combination of those parameters results in several indicators that are useful to determine the stress condition to which animals are exposed. The most widely used is the temperature–humidity index (THI), which combines air temperature with humidity providing a single index that gives information on animal comfort. The temperature–humidity index is mostly used for the measurement of HS condition because data relating to other parameters that could affect HS, such as solar radiation, wind speed, and precipitation, are not always available.

Numerous studies have established THI thresholds for HS in cattle but there are few studies that indicate the optimal values of THI for buffalo. A value of THI < 72 is considered optimal, THI between 72 to 79 is considered as mild stress, 80–89 is considered as moderate stress, and ≥90 is considered as severe stress in buffalo [34]. When the maximum THI increases above 72, mild stress conditions in buffalo are reflected in a slight decline in milk yield [34].

Considering the effects of THI on the reproductive performance of buffalo, Dash et al. [35] classified the months of the year into two categories based on the THI values: a non-heat stress zone (NHSZ) and a heat stress zone (HSZ). The months from October to March were included in the NHSZ with the THI ranging from 56.71 to 73.21, and the months from April to September were considered in the HSZ with the THI ranging from 75.39 to 81.60. Within the HSZ, a critical heat stress zone (CHSZ) was also characterized. The CHSZ corresponded to the months of May and June and the THI ranged from 80.27 to 81.60. Choudhary and Sirohi [34], based on an analysis of their data set, reported the maximum, minimum, and average THI threshold levels for buffalo: the cut-off values were 74.37, 61.73, and 68.15, respectively. The threshold level indicates the critical level of THI up to which the animal can tolerate HS and after which there is a significant productivity decline. Thus, in buffalo, although a decrease in milk production was perceptible after the maximum THI exceeded 72, the decrease was significant only when the THI exceeded 74.

Under HS conditions, buffalos modify various physiological parameters, such as their rectal temperature (RT), respiratory rate (RR), heart rate (HR), skin temperature (ST), and body temperature (BT) [36]; these are considered the main physiological parameters that can be quantified to evaluate the presence of stressful conditions, such as HS, in animals [22]. It has been observed that buffalo did not modify their BT, HR, and RR compared to cattle in barns when the air temperature ranged from 30–33 °C with low (33–38%) and high (82–88%) humidity [32]. However, if buffalos were in the scorching sun for a long time, the BT, HR, RR, and general discomfort of buffalo increased more rapidly than those of cows [32]. Gudev et al. [37] reported that the exposure of lactating buffalo to direct solar radiation (THI = 77.83) causes a significant increase in their RT and RR, showing that the heat load is greater than the body’s heat dissipation capacity. The same THI value did not induce significant changes in RT when buffalos were placed in the shade; although, maintenance of RT within the thermoneutral zone was achieved at the expense of a greater RR. A study carried out in Egypt [38] on buffalos and cows reported that under the same rearing conditions (the air temperature was 30–33 °C), the mean BT of both species was the same (38.8 °C), while the mean RR was widely different (27.19 for the buffalo and the 46.69 for cows for respiratory acts for minute). Those authors reported that between 5 and 7 years of age, the RT significantly decreases by 0.90 °C in buffalo and 0.25 °C in cows and remains constant thereafter. The respiration rate also follows the same pattern as BT, with it tending to decrease with advancing age.

The symptoms of the buffalo at different THI ranges were reported by Dash et al. [35]: with THI values below 72, there was no stress condition and the buffalos were in an optimal condition for reproduction and production; for the THI range between 72–78, there was a mild stress condition and there was an increase in RT and RR; for the THI range between 79–88, there was a moderate stress condition, RR and water intake were significantly increased, dry matter intake was decreased, and the ratio of forage to concentrate intake was decreased; and for the THI range between 89–98, there was a severe stress condition. In this case, excessive panting and restlessness were observed. Rumination and urination were lowered along with a negative impact on reproductive performance in the buffalo; a value of THI above 98 is an extreme stress condition and buffalo may also die.

From the studies examined, due to their peculiar physical characteristics, buffalo suffers from HS, especially in tropical areas. The threshold levels of the THI are higher than those reported for cows. Once these levels are exceeded, they react by modifying some clinical, reproductive, and production parameters.

## 3. The Effect of Heat Stress on Buffalo

### 3.1. Clinical Parameters

An overview of the effects of HS on physiological responses is presented in Table 1. All of the references agree that HS negatively affects RR and ST. About HR and RT, contrasting results are reported in the literature.

#### 3.1.1. Respiration Rate—RR

All of the references [27,39,40,41,42,43,44,45,46,47,48] agree that there was an increase in RR with the increase in THI. This result could be due to the increase in oxygen demand from the tissues under stressful conditions to try to maintain homeothermy by dissipating heat. This parameter seems to be the easiest and cheapest marker to evaluate HS levels. The interesting thing that we can note from Figure 1 is the big difference between the values recorded by the authors. As we can see in Figure 1, Li et al. [39] and Singh et al. [43] observed less than 20 breaths/min per season. Kumar and Kumar [41] found similar values in the morning in all of the seasons; however, in the afternoon of the hot–dry and hot–humid seasons, the RR rose between 40 and 50 breaths/min. Similar values have recorded by several authors [27,40,45,47] who observed an increase in RR from May to June and a decrease in RR from June to July despite the increase in the THI. A more evident RR increase was recorded by Shenhe et al. [42] and Wankar et al. [48]. Both studies observed less than 20 breaths/min at low temperature, but with the increase in THI, there was a significant increase in the RR which increased to about 70 breaths/min. Shenhe et al. [42] also examined the RR in the Mediterranean breed (MB) and a crossbreed (CB, Nili-ravi × Murrah) during the summer and observed that the MB is less tolerant to HS than the CB (76.84 ± 4.31 breaths/min vs. 60.82 ± 5.45 breaths/min, respectively). Wang et al. [46] found similar values for RR during the HS period but also a high RR during the non-HS period. The respiration rate peak, shown as a panting mechanism, corresponds to the incapacity of the buffalo to dissipate heat during the HS period.

#### 3.1.2. Heart Rate—HR

Regarding this parameter, a complete picture of the situation is presented in Figure 2. In the study of Chaudhary et al. [45], there was not a significant change in HR during the comfort zone, hot–dry season, and hot–humid season. In contrast, Talukdar et al. [40] and Manjari et al. [47] found an increase in HR during summer. This rise could depend on the increase in blood flow to the surface to facilitate heat loss. During winter, Manjari et al. [47] observed higher values for HR compared to Talukdar et al. [40]. Wankar et al. [48] registered a different trend. At first, there was an increase in HR passing through 25 °C to 30 °C. After that, at 35 °C, HR declined and then increased again at 40 °C. However, Singh et al. [43] found that HR was higher in the winter compared to the summer. Moreover, Yadav et al. [27] also observed that the higher THI corresponded to the lower values in HR. In general, Wankar et al. [48] and Yadav et al. [27] registered HRs between 35 and 55 beats/min despite the other authors [40,43,45,47] registering values between 60 and 80 beats/min for comparable THI values. The data from the literature clearly demonstrate the different responses of the buffalo to HS (Figure 2); this can be attributed to the different breeds (Murrah [27,40,43]; Surti [45] and Tarai [47]) or the different physiological phases of the animals in the studies (heifers of 9–13 months [40], heifers of 2 years [47], and heifers of 18–24 months [43]; lactating buffalo [27]; 4–5th lactation [45]; and dry buffalo of 9 years of age [48]).

#### 3.1.3. Rectal Temperature—RT

About RT (Figure 3), three studies [39,45,46] agree that there were no significant differences at different THI levels. In all other studies [27,40,41,42,43,44,47,48], a significant increase in RT was observed when the THI increased (Figure 3). Shenhe et al. [42] also found that CB buffalo (Nili-ravi × Murrah) had a lower value RT (39.12 ± 0.09 °C) compared to the MB (39.38 ± 0.09 °C) meaning that the CB buffalo are more heat tolerant. The change in RT is linked to major blood flow from the body core to the surface with the goal of dissipating endogenous heat from the skin in an attempt to restore thermal balance.

#### 3.1.4. Skin Temperature—ST

The increase in ST is a consequence of the pathway for the heat dissipation from the body core to the skin surface. All of the references [39,41,42,46] agree that during the exposure to HS, there was a significant increase in ST. The results are provided in Figure 4. Shenhe et al. [42] also examined the HS sensibility of two buffalo breeds. They compared MB to CB (Nili-Ravi × Murrah) and found that during summer, the MB showed higher value STs compared to the CB. Other authors [41] observed changes in physiological parameters between the seasons (spring, hot–dry HD, and hot–humid HU) and between morning and afternoon. The highest value of the THI was registered during the morning in the HU season and during the afternoon in both the HD and HU seasons which effectively corresponded to the greater value of ST. Thus, it is clear that with the increase in the THI, there was an increase in ST, but it is difficult to define a common THI threshold value and a specific THI breakpoint above which ST starts to increase.

Most of the clinical parameters considered in this review showed the same trend in the different buffalo breeds exposed to HS conditions. However, some crossbreeds seem more tolerant to HS (Nili-ravi × Murrah). Heart rate shows conflicting results across studies probably due to factors related to the individual studies (the breed monitored, physiological phase, age, etc.)

### 3.2. Dry Matter and Water Intake

Regarding dry matter intake (DMI), several authors [46,49,50] reported no effect of HS, even though Wang et al. [46] observed a slight reduction in summer. In contrast, other authors [51,52,53] have reported the significant negative effects of HS on the DMI (Table 2). This decrease could depend on the inhibition of the lateral appetite center in the hypothalamus by HS and by the fact that rumen fermentation produces heat which must be dissipated.

As shown in Table 2, exposure to HS conditions leads to an increase in water intake (WI) due to the physiological role of water in the organism, namely cooling of the reticulo-rumen and the dissipation of heat by sweating and panting. Sharma et al. [49] observed that WI can vary from 10.0 to 87.5 l/d with a mean of 29.9 ± 1.3 and 46.7 ± 3.1 l/d during winter and summer, respectively. Ashour et al. [51] and Wankar et al. [48] agreed with these results.

**Table 2 animals-13-01260-t002:** Effect of heat stress on dry matter intake (DMI) and water intake (WI).

Parameters	Effect of Heat Stress	Reference	Breed
DMI (kg)	↓**	[53]	Murrah
	↓*	[52]	
	↓*	[51]	
	n.e.	[46]	
	n.e.	[49]	Murrah
	n.e.	[50]	Murrah
WI (L)	↑*	[48]	
	↑*	[51]	
	↑*	[49]	Murrah

n.e. = no effect; ↑ = increase; ↓ = decrease; * *p* < 0.05; ** *p* < 0.01.

### 3.3. Reproduction

#### 3.3.1. Male

An overview of the effects of HS on semen parameters is presented in Table 3. Each parameter is analyzed below. No effect by HS on ejaculate volume was observed in all of the studies. For the other parameters, the response to HS was not very clear since different results were reported.

##### Ejaculate Volume (mL)

All of the studies reported no effect of HS on ejaculate volume. None of them showed a significant change during the seasons (Figure 5). Silva et al. [55] compared the ejaculate volume from eleven Murrah buffalo bulls during the period with the most rainfall (THI= 76.3 ± 3.6) and the transitional period (THI= 75.9 ± 3.5), and it was less in the rainy period (THI= 76.5 ± 2.8), but they did not find any significant differences, even though in the less rainy period, the volume was slightly higher compared with the other seasons. Ram et al. [54] analyzed 48 semen ejaculates from six Murrah buffalo bulls during the winter period (the temperature ranged from 15 °C to 25 °C) and the summer period (the temperature ranged from 42 °C to 48 °C). Even if the volume was higher in the winter compared to the summer, this difference was not statistically significant. On the contrary Ramadan et al. [57] showed that the ejaculate volume of four Egyptian buffalo bulls was lower in the autumn (the temperature ranged from 20.0 °C to 26.3 °C, THI = 70.6) and higher in the summer (the temperature ranged from 25.1 °C to 27.7 °C, THI = 74.9). Moreover, Tiwari et al. [56] and Sharma et al. [58] did not find differences between the seasons or THI classes, respectively (Figure 5). In particular, Sharma et al. [58] collected 41 semen ejaculated from 2 buffalo bulls; the experimental period was from April to October and was subdivided into three THI classes: THI < 72, THI between 72 and 78, and THI between 78 and 84. The volume recorded was a little bit higher in the first class (3.05 mL) compared with the other two (3.03 and 3.00, respectively), but this difference was not statistically significant. It can be concluded that the volume ejaculated ranged from 2.0 to 4.5 mL, and it was not affected by the HS conditions or season (Figure 5).

##### Sperm Concentration (Million/mL)

About sperm concentration, the effect of HS is not very clear (Table 3). As shown in Figure 6, Ram et al. [54] and Silva et al. [55] observed no significant effect of the season on the sperm concentration, even if, in both studies, higher values during the winter and the period with the most rainfall were observed compared with the less rainy period. In addition, Gonçalves et al. [59] reported the effects of two days of solar insulation on the sperm concentration of five Murrah buffalo bulls and observed an oscillation from 500.00 to 2500.00 million/mL but without any significant effect of solar insulation.

On the other hand, Sharma et al. [60] and Ramadan et al. [57] observed some significant differences between the seasons. Sharma et al. [60] found that when the THI was <72, the sperm concentration was higher compared to the conditions in which the THI was between 78 and 84. Similarly, Ramadan et al. [57] found that there was a significant decrease in sperm concentration in the samples collected during summer and autumn compared with the winter and spring samples. Moreover, a negative correlation (r = −0.483, *p* < 0.01) between the THI and sperm concentration was observed. In both of these two studies, it was shown that when the THI was over 70, the sperm concentration significantly decreased.

**Figure 6 animals-13-01260-f006:**
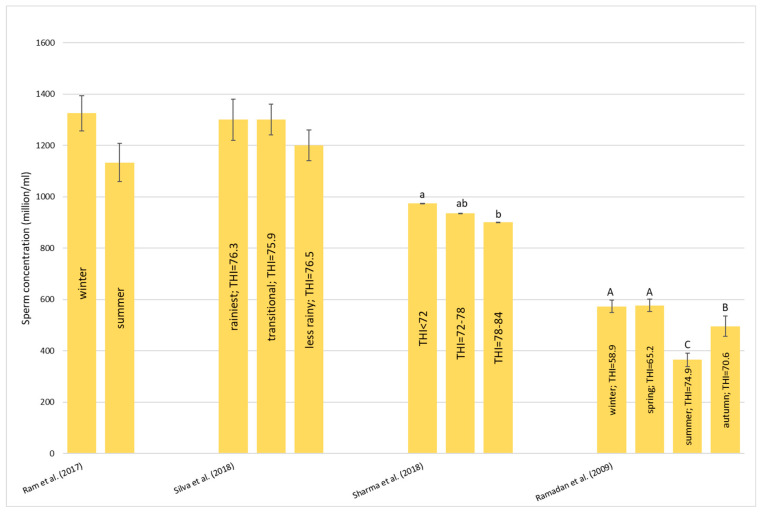
Values of sperm concentration (Mean ± SD) reported in the cited literature by several authors. ^a,b^
*p* < 0.05 ^A,B,C^
*p* < 0.01. Ram et al. (2017) [54]; Silva et al. (2018) [55]; Sharma et al. (2018) [60]; Ramadan et al. (2009) [57].

##### Total Number of Sperm Per Ejaculate and Mass Motility (0–5 Scale)

These two aspects were not investigated too much. Both Ram et al. [54] and Tiwari et al. [56] reported that the total number of sperm per ejaculate was not different between the seasons in the Murrah buffalo bulls. The values recorded were different in the two studies. Ram et al. [54] reported per each ejaculate 6575.74 ± 1306.37 million of spermatozoa in the winter (the temperature ranged between 15 °C to 25 °C) and 4785.31 ± 619.18 million of spermatozoa in the summer (the temperature ranged between 42°C to 48 °C), and Tiwari et al. [56] registered 3408.17 ± 49.70 million of spermatozoa in the winter and 3527.76 ± 51.90 million of spermatozoa in the summer.

About the mass motility on a scale from 0 to 5, both Ram et al. [54] and Sharma et al. [58] reported a significant decrease during the summer. Ram et al. [54] showed that mass motility was 3.63 ± 0.09 in the winter (the temperature ranged between 15 °C to 25 °C) and 3.33 ± 0.10 in the summer (the temperature ranged between 42 °C to 48 °C). Sharma et al. [58] showed that the mass motility was equal to 3.93, 4.07, and 3.67 with the THI below 72, between 72 and 78, and over 78, respectively. Even if there was a little increase with the increase in the THI, a significant difference was observed only between the THI from 72 and 78 and the THI > 78.

##### Initial Progressive Motility (%)

This type of motility refers to sperm that swim progressively, mostly in a straight line or in large circles. The results from the literature are reported in Figure 7.

Similar results were recorded by Ram et al. [54], who did not find any significant differences between the sperm samples in the winter and in summer, and Silva et al. [55] who did not find any differences between the period with the most rainfall, the transitional period, and the less rainy period. Moreover, Gonçalves et al. [59] did not find any significant differences pre and post two days of solar insulation. On the contrary, Ramadan et al. [57] found a significant increase in motility during the autumn compared to other seasons. Tiwari et al. [56] found lower values in the summer. The different results reported in the literature might be due to the different breeds (Egyptian or Murrah), the degree of sexual excitement, and the frequency of sperm collection and its method of collection [61]. Going forward, Sharma et al. [58] observed a significant reduction in progressive motility when the THI was between 78 and 84 compared to the motility recorded when the THI was between 72 and 78.

**Figure 7 animals-13-01260-f007:**
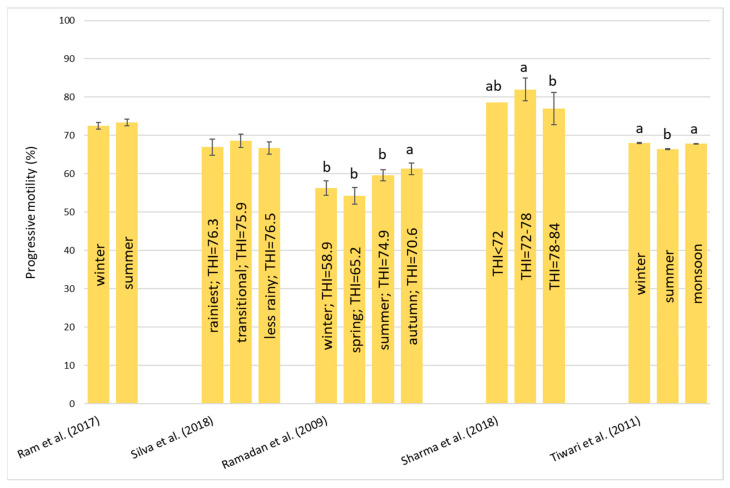
Values of progressive motility (Mean ± SD) reported in the cited literature by several authors. ^a, b^ *p* < 0.05. Ram et al. (2017) [54]; Silva et al. (2018) [55]; Ramadan et al. (2009) [57]; Sharma et al. (2018) [58]; Tiwari et al. (2011) [56].

##### Live Sperm Cells (%)

About the percentage of live sperm cells shown in Figure 8, it is not clear what happened during HS. According to Ram et al. [54], there was no significant difference between the seasons, and Gonçalves et al. [59] observed that there was no change in the percentage after two days of solar insulation. Instead, Ramadan et al. [57] found that there was a significant decrease in the number of live sperm cells during the spring compared to the other seasons. On the contrary, Sharma et al. [60] found that the percentage of live sperm cells significantly decreased with the increase in the THI. It is interesting to note that the values recorded by Ramadan et al. [57] for the Egyptian breed are generally lower than the values of the other two studies on the Murrah breed.

##### Abnormal Sperm (%)

In Figure 9 the presence of abnormal sperm is reported, and Silva et al. [55] noted that between the period with the most rainfall, the transitional period, and the less rainy period, there was not any significant difference. On the contrary, Gonçalves et al. [59] reported a significant increase in cell defects after two days of solar insulation. The first category was tail abnormalities which occurred between seven and fourteen days after insulation, followed by cytoplasmic droplets which were found between fourteen and twenty-eight days after insulation, and head defects found twenty-eight days after insulation. Thirty-five days after insulation, the defects did not differ from the values observed during the control period. Other studies have reported a highly significant difference between the seasons (Figure 9). Both Ram et al. [54] and Ramadan et al. [57] reported lower abnormal sperm percentages in the winter compared to the summer. Ramadan et al. [57] also found a positive correlation between abnormal sperm and the THI (r = 0.436, *p* < 0.01).

It seems clear enough that high temperatures damage testicular function which corresponds to a high percentage of sperm produced with abnormal morphology.

##### Intact Acrosome (%)

About the acrosome integrity of the semen, the two referenced studies disagree. Ram et al. [54] found that the percentage of intact acrosome did not change between the seasons. In contrast, Gonçalves et al. [59] found that after solar insulation for two days, there was a significant decrease in intact acrosome on day 28 after insulation.

##### HOST (%)

The HOST (hypo hosmotic swelling test) is a test to evaluate the integrity of sperm’s plasma membrane which is linked with the potential fertility of sperm. A high value in this parameter corresponds to the good quality of the semen. Ram et al. [54] found that the results for the HOST significantly decreased during the summer (52.08 ± 0.34% with the temperature ranging between 42 °C to 48 °C) compared with the winter season (55.88 ± 2.65% with the temperature ranging between 15 °C to 25 °C). Instead, Sharma et al. [60] found no significant differences between three the THI classes. The values registered were 76.33 ± 3.38, 73.33 ± 3.44, and 72.00 ± 4.76% for THI < 72, between 72 and 78, and between 78 and 84, respectively. The values recorded by the second authors are higher than the other, but the breed, the age of the bulls, and the method used to collect the samples were the same.

#### 3.3.2. Female

Buffalos are polyestral animals capable of reproducing throughout the year; in any case, a seasonal trend is recognized in many regions of the globe [62]. Sexual activity is increased with the decrease in the day length and temperature [63] resulting in the highest breeding frequency in the winter and the lowest frequency in the summer. Seasonality is clearly linked to nutrition; generally, births occur at a time of greater availability of feed. Even in Italian latitudes, the seasonality of this species is present, even though a constant and good quality diet is provided all year round. This is due to the photoperiod, which, by acting on the release of melatonin, can influence the fertility of the buffalo as much as feeding [64].

The reproductive characteristics of buffalo females have a very variable range due to the enormity of the breeds present. Buffalo is characterized by a high age at puberty which ranges from 16 to 46 months, with it being 30 months on average [62], a body weight between 200–350 kg, generally 55–60% of adult body weight, and the length of estrus cycle varies from 17 to 26 days, with 21 days being the average, and the length of estrus varies from 5 to 27 h. The length of gestation varies between the river and swamp type, with it being around 300–320 days for the first type and 320–340 days for the second type. The birth weight of the calves varies between 22–36 kg; the days required for uterine involution are between 25 and 35 days [62]. The reproductive efficiency of buffalo is affected by endogenous factors such as hormones and exogenous factors such as climate, nutrition, disease, and management. The temperature–humidity index plays an important role in the reproductive functions of buffalo, and several authors have suggested that a THI > 75 has a negative effect on the reproductive performance of buffalo in tropical zones [65,66]. An overview of the effect of HS on female reproduction is presented in Table 4.

##### Estrus: Its Cycle Length and Expression

In river buffalo, the average length of the estrus cycle is 20–22 days [67]. A seasonal variation was reported wherein the duration of estrus was estimated to be 18 h and 8–10 h in the winter and summer seasons, respectively, in river buffalo. In addition, for Murrah buffalo, Upadhyay et al. [31] reported that estrus expression was reduced during the summer months.

Under hot climates, the duration of estrus tends to be shorter [62,66]; the signs may be exhibited in the coolest hours of the day such as in the morning (6.00 h) and midnight (24.00 h) and not at noon time (12.00 h) [22]. This, therefore, leads to low expression of heat or silent heat, which it is one of the deleterious features to the reproductive performance of buffalo. The poor expression of symptoms of estrus, especially during the hot summer months, leads to a lengthening of the calving interval and, therefore, to poor reproductive and productive efficiency [66].

The period of postpartum anestrus or acyclicity is highly variable in buffalo in that can resume cyclicity by 30–90 days [62]. Anestrus has a variable incidence between 36.6–59.5%, but during the summer season, it tends to increase [80]. Animals that calved in the spring and summer had a higher incidence of anestrus compared with those that calved in the winter season [81].

##### Pregnancy Rate—PR (%)

About this parameter, all the studies agree: HS can negatively affect the PR in three different ways: i) in the establishment of pregnancy; ii) it is associated with pregnancy failure; iii) in late pregnancy or the postpartum period [66]. A reduction in PRs in the summer compared to the winter has been reported by many authors. Megahed et al. [69] observed a reduction in PRs from 90% (in the winter season) to 62.5% (in the summer season). Dash et al. [70] reported that the threshold value of the THI above which negative effects occurs was 75 in Murrah buffalo. In fact, a reduction from 41 to 25% was found covering the range from THI 70–74.99 to THI 75–79.99. In another study, Dash et al. [65] observed a greater reduction in PRs in buffalo at the first parity, with the rate going from 46% when the THI ranged from 70 to 74.9 to 24% with the THI ≥ 75.

One cause of PR reduction during summer HS can be attributed to the effects on oocyte quality [69]; this will be explained in more detail.

##### Quality of Oocytes (%)

In addition, the quality of oocytes is influenced by HS. Zoheir et al. [71] found a low percentage of good-quality oocytes in the summer months vs. the winter months (50.0 vs. 74.6%, respectively); fair oocytes were more present during the spring months, and the number of denuded oocytes was higher during the summer months. The maturation rate was also consequently lower in the summer months and higher in the winter months (59 vs. 92%, respectively). A recent study carried out in Bangladesh [72] confirmed that during the summer season, the quality of oocytes was very low compared with the winter season, and the average surface follicle count per ovary was significantly higher in ovaries collected during the winter season than in the summer [72].

The cause of the greater presence of poor-quality oocytes in the summer could be derived from the effect that HS exerts on hormonal balance by reducing follicular development [82].

Di Francesco et al. [73] reported in Mediterranean Italian buffalo that no effect of the hot season on oocyte quality occurred, but on the contrary, the percentage of degenerated oocytes was lower in the summer season (18.1 ± 1.5% in July–September, with the mean ambient temperature of 23.3 ± 0.5 °C), compared to the winter season (23.0 ± 1.1% in October–December, with the mean ambient temperature of 13.3 ± 0.7 °C).

Variations in oocyte quality between the studies could be attributed to differences in breeds, agroclimatic conditions, the physiological phase of the animals, the season of ovary collection, the number of ovaries processed, the techniques used, and the criterion for selecting ovaries [72].

##### Days Open—DO (days)

Regarding DO, contrasting results can be observed in the literature. Aziz et al. [75] reported an increase in DO in the winter season compared to the summer season (190.73 ± 5.37 vs. 165.59 ± 5.11 days, respectively). Aziz’s study was based on 13-year data (1979–1992) from Egyptian herds. Nasr [74] in a study evaluating the HS sensitivity of three different breeds found that the pure Egyptian (PE) breed and a crossbreed (Egyptian buffalo 50% × Mediterranean Italian buffalo 50%) revealed a significant increase in days open (8 and 29 days, respectively), from a low to high THI (≤ 70 and over 80, respectively). Instead, for the crossbreed (Egyptian buffalo 75% × Mediterranean Italian buffalo 25%), there were no effects on DO by the THI.

##### Lactation Length—LL (days)

Lactation length (LL) is affected by several factor including parity, management, health, and calving season [74,75,77]. Thiruvenkadan et al. [77] reported that Murrah buffalo that calved during the summer season showed a trend to a reduction in LL compared with the winter season (293.3 ± 5.0 vs. 301.8 ± 3.7 days for the summer and winter seasons, respectively). Those authors also showed that as the parity increases, the LL decreases. Aziz et al. [75] reported that for the first parity, LL was shorter compared to the other parameters. However, in a retrospective analysis of 13 years, Aziz et al. [75] observed that LL was unaffected by the season of calving. Soysal et al. [78] described that LL was lower in the first parity, was constant from the second to the fourth parity, and then decreased, and the hot season negatively affected LL, reducing it by 29 days compared to the cold season in Anatolian buffalo (244.82 vs. 215.80 days for the winter and summer seasons, respectively). On Anatolian buffalo, Alkoyak and Öz [79] reported a negative effect for the hot season (in winter, 265.35 ± 4.09, and in summer, 251.68 ± 3.29), but instead, there was no difference between parity.

On the contrary, the comparison of heat sensitivity made by Nasr [74] on genetic types of buffalo found that crossings with the Italian × Egyptian buffalo tended to increase the LL (29 and 46 days for BC (75% Egyptian × 25% Italian) and F1 (50% Egyptian × 50% Italian), respectively) from a low to high THI (≤70 and over 80, respectively). Instead, pure Egyptian did not show any changes in LL. Furthermore, Jakhar et al. [76] did not report any effect on LL of the winter and summer seasons on Murrah buffalo but nevertheless reported a significant effect of parity, confirming what was reported by other authors.

##### Calving Interval—CI (days) and Dry Periods—DP (days) Length

Generally, buffalos have very high CIs, ranging from 471 to 585 days [81]. The optimal CI for dairy buffalo should be 12–13 months [81]. Alkoyak and Öz [79] found that in Anatolian buffalo, the CI decreased in the summer and spring seasons by about 28 days compared to the winter season (390.28 ± 8.57, 390.62 ± 7.50, and 418.04 ± 9.34 days for the spring, summer, and winter seasons, respectively). Jakhar et al. [76] observed a seasonal effect on the CI; it was lower in the summer and rainy season than in the winter for Murrah buffalo (463.73 ± 5.63, 471.43 ± 9.66, and 504.61 ± 6.24 days for the rainy, summer, and winter seasons, respectively). On the contrary, Thiruvenkadan et al. [77] reported, in Murrah buffalo, that the CI did not differ between the summer and winter seasons, but there was a marked parity effect: the CI tended to decrease from the first to the second parity. In addition, in Anatolian buffalo, the season did not have any effect on the CI [78]. Nasr [74] observed that in PE buffalo and an F1 crossbreed (Egyptian buffalo 50% × Mediterranean Italian buffalo 50%), the CI increased by 9 and 29 days, respectively, from a low to high THI (≤70 and over 80, respectively), while the BC genotype (Egyptian buffalo 75% × Mediterranean Italian buffalo 25%) was not affected by the THI.

There are not many studies on the effects of HS on the DP. Thiruvenkadan et al. [77], investigating Murrah buffalo, did not find any influence by season on DP length; on the contrary, Jakhar et al. [76], investigating Murrah buffalo, reported that winter-calved buffalo had a longer DP than summer-calved ones (190.33 ± 7.29 vs. 173.75 ± 11.47 days for the winter and summer seasons, respectively), and they also reported that there was a significant influence of parity on the DP. In addition, Aziz et al. [75], investigating Egyptian buffalo, found that buffalo calved in the winter had a longer DP compared with buffalo calved in the summer (279.66 ± 6.34 vs. 259.75 ± 6.03 days for the winter and summer season, respectively).

##### Conception Rate—CR (%)

Conception rate is an important reproductive trait because if the average CR is reduced, the service life and calving interval increase leading to economic losses at the farm level. About the CR, all of the studies reviewed agreed that high temperatures tend to reduce CRs in buffalo (Table 4). Upadhyay et al. [31] reported a reduction in CRs with a high THI (32.97% with 81.22 THI in June); the highest value CR was registered in April (47.34% with THI = 75.72).

Dash et al. [70], in a 20-year retrospective study on Murrah buffalo (1993–2012), reported that CRs are strongly influenced by the THI. When the THI exceeded the 75 threshold, the CR was reduced by 9 percentage points (76 vs. 67% CR when the THI exceeded 75).

Nasr [74] found that the genotype BC (Egyptian buffalo 75% × Mediterranean Italian buffalo 25%) had the highest CR compared to PE and F1 (Egyptian buffalo 50% × Mediterranean Italian buffalo 50%) at all levels of the THI. Increasing the THI reduced the CR percentage in all three crossings; the CR was reduced by 4%, 10%, and 21% for PE, BC, and F1, respectively, from a low (≤ 70) to high (> 80) THI.

##### Service Period—SP (days)

Service period is a parameter that has not been investigated much; only a few studies have taken it into consideration. The average SP of Murrah buffalo was prolonged (180 days) in May with the corresponding THI value of 80.27. On the contrary, the lowest average SP (119 days) was observed in the month of March at an average THI of 67.80 [35]. The service period in a study by Dash et al. [70] was reduced when the THI exceeded the value of 75. In this study, the average SP was 127 days for the THI subclass 70.00–74.99, and it increased to 162 days in the THI subclass 75.00–79.99. While in the study by Jakhar et al. [76] on Murrah buffalo, on the other hand, no effect of the winter and summer seasons was evident.

In conclusion, regarding the reproductive traits of buffalo males, the most investigated breed is the Murrah. Some reproductive parameters (such as ejaculate volume, total number of sperm per ejaculate, and a part of study on sperm concentration, initial progressive motility, and live sperm) are not affected by HS, and for other parameters (mass motility, abnormal sperm, intact acrosome, and HOST), the negative effects of exposure to hot conditions are marked so much that they compromise the reproductive efficiency of the males suffering from HS. For the reproductive aspects of females, there are many published studies, with many parameters investigated on different breeds and crossbreeds in different physiological phases and parity. It is clear that HS affects multiple reproductive traits of female buffalo. Under hot conditions, estrus decreases, and silent heats increase; furthermore, the service period increases. Conception rates and pregnant rates are reduced, especially in the first parity, and this could be due to a low percentage of good-quality oocytes due to hormonal imbalances by reducing follicular development. The effects of HS on the number of days open, the lactation length, and calving interval are not clear enough and are sometimes conflicting between the different studies. This is probably due to parity, breed, health, nutrition, and herd management. Further studies to better investigate these parameters are needed.

### 3.4. Milk Yield

Milk yield (MY) is affected by several factors: i) physiological, which will be governed by the genetic make-up of the buffalo and ii) environmental, such as age, the number of lactations, days of lactation, lactation phase, season, pregnancy, season of calving, calving interval, and nutritional and health status [13,83,84]. Most studies have dealt with the problem of HS on buffalo reared in areas with a tropical and/or sub-tropical climate [31,34,85,86,87]; however, few studies have been carried out on Mediterranean buffalo breed in temperate climates. The reported results are conflicting because they refer to different breeds of buffalo, different climates, and different breeding conditions. An overview of the effect of HS on MY is presented in Table 5.

De Rosa et al. [88] observed lower MY in Mediterranean Italian buffalo exposed to hot conditions. In contrast, other authors [23,68,84] did not report any effects of the hot season on MY in Mediterranean Italian buffalo. This is probably due to better adaptability of the buffalo to high temperatures [68]. It would rather appear that the Mediterranean buffalo breed is much more susceptible to low temperatures [23,68].

According to the report obtained from the data provided by the Tuscany Regional Breeders Association, concerning the production from 2007 to 2011 of 402 buffalo of the Mediterranean Italian breed, Salari et al. [84] observed that the annual MY was homogeneous throughout the year (9.42 kg/head in the winter, 8.94 in the spring, 9.37 in the summer, and 9.56 in the autumn). There were no differences between the summer and the other seasons, but there was a decrease in the spring season (−5.4%). The decrease in MY during the spring months could be due to the greater presence of buffalo at the end of lactation, and the lack of effects in the hot season on MY might be masked by the seasonality of parturition, concentrated at the beginning of the summer months due to market needs.

A two recent study carried out in southern Italy [23,68] argues that in the Italian climate, low temperatures influence MY negatively. Zicarelli [68] in a 4-year study (from 2017 to 2020) reported that when the ambient temperature decreases, there is an increase in the percentage of subjects with decreased MY. This author reported that during the coldest seasons (with an ambient temperature = 8.13 °C on average), the percentage of subjects that showed a decrease in MY was 41.22% against 29.91% of subjects during the hottest seasons (with an ambient temperature = 26.44 °C on average). Zicarelli [68] also reported that the phenomenon was more evident in pregnant females than in non-pregnant ones (46.83% vs. 36.03%, respectively). This study was carried out on a single large farm, considering all lactating animals. Furthermore, the animals were provided with swimming pools between May and October which allowed them to cool off. As reported in the present review, in Section 4.2, swimming pools are currently one of the best cooling systems for dairy buffalo.

Matera et al. [23] observed an unfavorable effect on MY when the THI was below 59.0, suggesting that low temperatures and humidity may have a detrimental effect on MY. On the contrary, when the THI was higher than 60.0, the MY was more stable. They suggested that this phenomenon might be attributed to the tropical origin of the species and to progressive adaptation after the stress period.

Buffalo raised in tropical and sub-tropical climate are negatively affected by high temperature. As reported before, high temperatures cause stress due to increased body heat which leads to low heat dissipation from the body surface. The high thermal load in lactating buffalo reduces MY and the duration of the lactation periods in Murrah buffalo [31,78,79]. Upadhyay et al. [31] observed that when the THI was above 80 (in the summer season), the lactation period of the buffalo was shortened (3–7 days). Furthermore, these authors reported that both cold and heat waves have a strong negative impact on the MY of buffalo and that these events persist for several days. The return to normal MY takes 2–5 days with a variable response in individual buffalo. Moreover, the same authors observed that the extent of decline in MY was lower at the mid-lactation stage than either the late or early stage. The decline in MY varied from 10 to 30% in the first lactation and 5–20% in the second or third lactation.

Pawar et al. [85] reporting on a 5-year data set of Murrah buffalo found that MY (kg for 305 days) in the winter (November–February) was significantly higher than in the summer (March–June). Those authors also found that parity had no effect on MY. During the summer season, the MY was 1951.6 ± 125.9 L/lactation, while during the winter, it was 2300.6 ± 108.3 L/lactation. This study also indicated that the buffalo that calved in the summer produced less than those that calved in the winter; this has already been widely seen in the dairy cows [19]. Pawar et al. [86] reported a decrease in MY from 4.46 to 3.65 kg as the THI increased from 74 to 83 during the summer in Murrah buffalo. They indicated that the yield drops by 0.028 kg per buffalo per day for each point increase in the value of the THI above 72. Choudhary and Sirohi [34] in a retrospective study (10 years) observed that the critical threshold level of the maximum THI was 74. The duration of the hardship period for buffalo begins in mid-March and lasts until the beginning of November. During the aggravated stress condition (THI > 82), the MY decreased by more than 1% per unit increase in the maximum THI above 82.

Nasr [87] evaluated the sensitivity to HS of three different breeds: Egyptian (PE); a F1 cross between Egyptian buffalo (50%) × Mediterranean Italian buffalo (50%); and a BC cross between Egyptian buffalo (75%) and Mediterranean Italian buffalo (25%). This author observed that both PE and BC were quite robust at high temperatures; PE only had a reduction in peak MY from 15.02 to 13.72 kg in subjects exposed to a THI ≤ 70 and a THI > 80, respectively. The BC showed the same trend as the PE; there was a reduction in peak production (from 15.52 to 14.04 kg for the low and high THIs, respectively) and a reduction in total production (2331.92 and 2327.50 kg/lactation for low and high THIs, respectively), while the F1 showed a statistically significant reduction in MY from 10.33 to 8.38 kg/day in buffalo exposed to a low THI (≤70) or a high THI (>80), respectively. This study demonstrates the greater adaptability of buffalo native to tropical climates to high temperatures; in fact, the PE and BC (100% and 75% Egyptian buffalo, respectively) showed a slight reduction in milk production at different THI levels. Instead, the F1 (50% Egyptian buffalo and 50% Mediterranean Italian buffalo) reduced their daily milk production by 18.88%, passing from a low level to a high level of the THI and showing greater sensitivity to HS due to the greater presence of Mediterranean Italian buffalo (50%).

### 3.5. Milk Composition and Characteristics

An overview of the effect of HS on milk composition and characteristics is presented in Table 6. All of the parameter results from the literature are discordant.

Only for Wang et al. [46], neither fat nor protein were affected by HS even if in both cases they registered a reduction in fat and protein percentages (Figure 10 and Figure 11). Instead, for fat percentage, most authors [13,30,84,85,86] observed a significant decrease in buffalo exposed to hot conditions (Figure 10). In particular, Pawar et al. [86] found that during the hot season, the fat percentage decreased by 0.047% per each point increase in the THI in Murrah buffalo. In addition, Matera et al. [23] showed that when the THI was above 69.0, the fat percentage was negatively affected. A negative effect was also observed when the THI was lower 47.0. These results indicate and confirm that buffalos are susceptible not only to high temperatures but even to lower temperatures. On the opposite side, Nasr [87] found that in all the genetic types considered in the study (PE, F1, and BC), the fat percentage increased passing through the THI < 70 to the THI > 80.

About the protein percentage (Figure 11), Pawar et al. [86], Wang et al. [46], and Nasr [87] reported no effect of HS. Other authors [13] showed a decrease in protein percentage in the spring mainly due to the lactation phase more than the effect of the season. The study of Salari et al. [84] reported lower concentrations of protein in the summer; likewise, Costa et al. [89] found a lower protein concentration with the increase in the THI. On the other hand, Matera et al. [23] showed that the THI has a positive influence on protein percentage but with a drop when the THI was around 62.

For lactose percentage, discordant results are reported. Nasr [87] found that there was not a significant difference in lactose percentage between the three THI classes considered and for all the genetic types tested. Costa et al. [89], in bulk milk obtained from two milkings, observed lower percentage with the increase in the THI. In particular, those authors found that lactose percentage was reduced from 4.80 to 4.79 and 4.73 in buffalo exposed to a THI = 39.66–47.72, a THI = 47.98–68.94, and a THI = 68.95–76.09, respectively. On the contrary, Pasquini et al. [13] showed a greater concentration of lactose in the summer (4.91 ± 0.06%) compared to the winter (4.59 ± 0.07%).

About milk urea nitrogen (MUN), Wang et al. [46] did not observe a significant change between the HS and non-HS periods. Costa et al. [89] found that the concentration of MUN increased significantly with the increase in the THI (THI = 39.66–47.72, MUN = 27.81 mg/dL; THI = 47.98–68.94, MUN = 33.61 mg/dL; THI = 68.95–76.09, MUN = 39.18 mg/dL).

Costa et al. [89] reported changes in other milk characteristics in bulk milk. Those authors observed an increase in the electrical conductivity (EC) of milk with the increase in the THI above 72. Those authors [89] also studied the effects of the THI on the pH and coagulation properties of milk and observed a reduction of rennet coagulation time (RCT) and an increase in curd firmness (a_30_) with the increase in the THI indicating better cheese making properties. These results did not agree with changes in fat and protein percentages that were significantly reduced when the THI increased. It has to be considered that milk clotting parameters are also influenced by many factors including the microbiological profile of the milk which is different between individual samples and bulk samples.

Regarding milk somatic cell score (SCS) two studies [46,89] found no effects of HS. Salari et al. [84] observed a significant reduction in SCS during the spring and this trend agreed with the findings by Pasquini et al. [13]. Matera et al. [23] found a greater SCS when the THI was lower than 51.0 or higher than 69.0, highlighting a susceptibility to mastitis either during heat or cold stress. As is well known, the SCS is a multifactorial trait and can be affected by parity, the stage of lactation, the season of calving, and management. Thus, it is not easy to deepen the relationship with the THI only.

Finally, for the total bacteria count (TBC), Pasquini et al. [13] did not observe a significant change between the seasons despite there being an increase in the summer. Costa et al. [89] showed an increase in TBC with the increase in the THI.

Regarding milk production, Mediterranean Italian buffalo showed an impairment in their milk quality when exposed to hot conditions, with little effect being observed for milk yield. The breed that showed the greatest loss of milk when exposed to HS conditions seems to be the Murrah breed.

## 4. Mitigation Strategies

A reduction in the deleterious effects of HS is possible through acting in different ways. One strategy is the genetic approach by selecting subjects that are more tolerant to HS. For example, the Mediterranean Italian breed seems to be less tolerant than the Egyptian breed and the crossbreed (Nili-ravi × Murrah) [42,87]. Other approaches are feeding, nutritional, and cooling strategies (Figure 12).

### 4.1. Feeding and Nutritional Strategies

Different nutritional strategies have been implemented (Table 7). Some of them [69,90,91] supplemented the diet with minerals and/or vitamins that act like antioxidants, i.e., chemical substances that counteract the production of free radicals formed during exposure to HS [92]. Wafa et al. [93] supplemented the diet with *Moringa oleifera* leaves (MOL) which contained tannins, glycosides, anthocyanin, thiocarbamates, and polyphenols that can protect against oxidative stress. Moreover, Lakhani et al. [94] and Talukdar et al. [40] tested different diets in which there were different levels of fiber, protein, and energy with the aim to test diets with different metabolic heat production potential in the rumen.

Evangelista et al. [90] and Chaudhary et al. [91] found that treatment did not affect dry matter intake but positively influenced MY with an increase in 3.5% and 10.9% L/day, respectively. Evangelista et al. [90] did not find any significant differences in milk fat, protein, and lactose content. In contrast, Chaudhary et al. [91] observed an increase in fat, protein, and lactose percentage. Moreover, Evangelista et al. [90] examined milk coagulation traits and observed the positive effects of treatment on milk coagulation properties. Chaudhary et al. [91] and Wafa et al. [93] found positive effects for treatment on physiological parameters showing a significant reduction in RT, ST, and RR (Table 7). Wafa et al. [93] also examined the effect of MOL on semen quality and reported a significant increase in the motility and viability of the semen and a decrease in the incidence of abnormality and damaged acrosome.

Other authors [69] fed the buffalo with supplementation of vit E + sodium selenite and observed that the pregnancy rate improved from 62.5% to 75.0% during the summer season.

Kumar et al. [95] administered an intra-muscular injection of vit E + Se and reported an increase of more than 3 kg/d in the DMI and an increase in MY of 1 kg/d and milk fat percentage (+0.5 percentage points) in the treated group compared to the control group. In addition, there was a decrease in RT (−1 °C), RR (−3 breaths/min), and HR (−5 beats/min) in the treated group compared to the control group.

These findings show and validate the importance of antioxidant supplementation during HS.

Lakhani et al. [94] found no effects of the different levels of fiber and proteins on the DMI and MY, but there was an improvement in milk quality. All of the diets increased the fat percentage, with the best value of 6.81 ± 0.05 for the diet containing 37% NDF and 8% metabolizable protein (MP) compared to the control diet containing 30% NDF and 7% MP. In addition, those authors observed a reduction in HR from 50 beats/min in the control diet to 46 beats/min in all the diets containing 8% MP. The treatment was also able to decrease the RT (about −0.2 °C) and RR (about −3 breaths/min). The best diet that improved the physiological parameters was that containing 34.5% NDF and 8% MP.

In addition, Talukdar et al. [40] investigated the effects of two different diets on physiological parameters and observed that increasing the metabolizable energy by 15% did not have any effect on RT, ST, and RR. In contrast, the reduction by 15% of metabolizable energy resulted in an increase in RT and ST; this was probably due to greater metabolic heat production derived from the increase in fiber content in the diet.

### 4.2. Cooling Strategies

A strategy to prevent HS in dairy buffalo is an intervention in their housing system and the adoption of cooling systems (Figure 12, Table 8). Several authors have ascertained the positive effect of cooling on the various parameters indicating the well-being of the animals in both temperate and tropical climates [88,96,97,98,99] and in both female and male buffalo [100]. These systems may affect several aspects of buffalo breeding including reproduction. Neglia et al. [96] observed in multiparous lactating Mediterranean Italian buffalo that a greater availability of space and the presence of a swimming pool improved the conception rate within 120 days post-partum (53.7% vs. 39.9% in the group with a pool and without a pool, respectively). Furthermore, the animals with a swimming pool showed a lower calving–conception interval than the group without a swimming pool, especially in the hottest months (April–August).

In the previous paragraphs, the negative effect of HS on males and the deterioration of sperm quality under hot conditions was discussed [57,59,60,88]; furthermore, the maximum temperature for optimum spermatogenesis was defined as 29.4 °C, whereas the minimum temperature was 15.5 °C [43]. When these thresholds are exceeded, the animals go into stress and spermatogenesis is negatively affected. Hoque et al. [100] found that using more showers a day can improve the quality of sperm. They tested the effect of multiple showers on sperm quality in males under controlled climatic chamber conditions (the THI medium was 72.66 ± 2.30 in the morning and 81.66 ± 2.51 in the afternoon). The treated group was given four showers per day and the control group was given only one shower. The treated group was affected positively for the ejaculated volume (mL), live sperm (%), and sperm concentration (million cells/mL). They also found a higher percentage of normal fraction in the treated group compared to the control.

The presence of a pool improves both the quantity and quality of the milk and improves the heat dissipation capacity of the animals. It has been seen that swimming seems to be a more beneficial method than showers [97]. The research carried out by Aggarwal and Singh [97] on the comparison of two types of cooling systems for buffalo at the beginning of lactation clearly indicated that the group with the swimming pool presented greater feed intake and MY during the hot–dry and hot–humid seasons (the THIs were 80.3 and 83.6 during the hot–dry and hot–humid seasons, respectively). Moreover, the RT and RR were lower in the wallowing group especially during the evening time.

De Rosa et al. [88] on multiparous buffalo (3.20 ± 0.25 number of lactation) reported that the pool had a positive effect on MY (an increase of 1.8 kg/head in the hot period, i.e., the month of July with an average daily temperature of 24 °C). The characteristics of the milk were not influenced (fat and protein content), but the milk from pool group showed a greater tendency for higher somatic cells. The behavior of the buffalo was also affected by the presence of a pool. The pool group showed greater socialization activity between individuals (sniffing and nuzzling) and more agonistic interactions compared with the group without a pool. Therefore, the animals without the possibility of having an effective cooling system were affected by HS which determined the reduction in potential milk production and reduced their overall well-being. The authors, in fact, argued that the effective heat dissipation through wallowing by the buffalo with a pool may have contributed to sustaining the buffalo MY.

Other researchers [27] compared the wallowing system with misting over three months period (May, June, and July and the average THI of these three months was 79.88, 80.57, and 85.36, respectively). Milk production decreased in both experimental groups, but the decrease was more evident in the misting group. Rectal temperatures and RR decreased in both groups compared with the control group but decreased more in the wallowing group in July. The heart rate increased in the wallowing group. Those authors concluded the wallowing and misting were equally effective in preventing a decline in MY during May and June (the hot–dry period); however, wallowing was highly effective during July (the hot–humid period) in maintaining MY.

Recently two similar studies, on Nili-Ravi buffalo, were carried out to test three different cooling methods; the first study was carried out during a hot–humid summer [98] and the second study was carried out during a hot–dry summer of a subtropical region [99]. The three systems adopted were: shade (S), shade with fans (SF), and shade, fans, and sprinklers (SFS). The results were similar in both climates. Greater MY was observed in the SF and SFS than in the control in both studies. The milk quality was better (fat, protein, and solid no fat) for the SF and SFS than the control, while no difference was found for the lactose content for the three groups in the humid climate [98]; instead, in the dry climate [99], the lactose content was also higher in the SF and SFS than in the S group. Under a hot–dry climate [99], a greater DMI in both of the treated groups (+17.34% and +5.40% for SFS and SF, respectively compared to the S group) was observed. Water intake was greater in the control group than in the two treated groups. The same results were also obtained during a hot–dry summer season [99]. In the hot–humid climate, several parameters were reduced such as the RR (reduced by 24.70 and 42.47% breaths per minute in the SF and SFS groups, respectively, compared to the control), the pulse rate (reduced by 10.36% and 23.32% beats per minute the for SF and SFS, respectively, compared to the control), and the ST (reduced from 0.87 in the SF and 2.09 °C in the SFS compared to the control).

Ahmad et al. [99] also investigated the effects of three cooling systems (S, SF, and SFS) on the feeding behavior of buffalo during a hot–dry summer climate. They found changes in eating activities. The eating time (min/24 h) was influenced by the treatments increasing from S to SFS (246.33, 280.33, and 309.50 min/days for S, SF, and SFS, respectively). The total time spent in drinking (min/24 h) the S, SF, and SFS was 24.67, 22.50 and 19.50, respectively. In addition, finally, the time spent in rumination was higher in the SFS (399.00 min/24 h) followed by the SF (385.17 min/24 h) and S (360.83 min/24 h). Furthermore, they observed that the total time spent in lying showed a greater value in the SFS than in the S. The time spent in locomotion (min/24 h) was 92.50, 76.33, and 68.67 for S, SF and SFS, respectively. All of these changes indicate greater well-being of the buffalo treated with SFS compared to the buffalo treated with SF and those treated with only S. Those authors concluded that the best cooling protocol to be used is the combination between shade, a fan, and sprinklers.

Recently, silvo-pastoral systems have been proposed as an alternative method to conventional breeding [101,102]. Athaide et al. [101] comparing two herds of buffalo reared one under a silvo-pastoral system (SPS) and the other one under a conventional system (CVS, pens with no shade) and found that subjects in the CVS had higher RT, ST, and HR. Galloso-Hernández et al. [102] carried out a study on heifers under two rearing systems: SPS and CVS. They reported that in the SPS, the ambient temperature was 2 °C lower than in the CVS (33.01 vs. 31.00 °C), and the feeding behavior (time spent eating, ruminating, and drinking water) was greater in the SPS under intense HS conditions (>75 THI). Under the CVS, the animals spent more time wallowing compared with the SPS. Galloso-Hernández et al. [102] concluded that the time animals spent for thermoregulation was greater in conventional systems than in the SPS.

## 5. Conclusions

Heat stress causes a huge loss in MY and alters milk characteristics in dairy buffalo. HS also reduces the DMI which is directly co-responsible for the lower MY. The reduction in the DMI due to the increase in the THI is a known strategy to reduce the heat load that reduces the production of endogenous heat in an animal’s body which is produced by digestive and metabolic processes. Moreover, HS, in most cases, has an impact on milk quality through a reduction in fat, protein, and lactose and an increase in MUN, TBC, and SCC. Heat stress negatively affects physiological condition by increasing RR, HR, RT, and ST and reduces reproduction efficiency in both males and females by altering the quality of semen and reducing conception rates and the quality of oocytes, respectively. In males, even if HS does not affect ejaculate volumes, it causes a decrease in fertility due to its negative effect on sperm concentration, motility, viability, and defected spermatozoa. In females, HS causes difficulties in the induction of heat, a reduced lactation length, lower pregnancy rates, bad-quality oocytes, and an increase in the service period and calving interval.

Different approaches are used to mitigate the negative effects of HS on buffalo. Feeding management and the integration of diet with antioxidant substances can help the animal in fighting HS. In addition, the modification of their diet, with particular attention to fiber concentration and quality and protein quality, can be a useful strategy to mitigate the negative effects of HS.

Cooling is the first thing to carry out to keep buffalo cool. Between the cooling systems, the use of pools is the most effective. Wallowing can prevent increases in RT, RR, and HR by decreasing the heat load of the body by enforcing additional physical heat loss from the body. Wallowing also improves DMI, MY, and milk quality and the eating and social behavior of buffalo which, through wallowing, can demonstrate its typical species-specific behavior. Wallowing is more effective because besides evaporative heat loss, conductive and convective heat loss also prevail during wallowing. This is very important in buffalo because the low hair density on their skin helps in readily exchanging heat from their skin to water. In addition, the use of fans and sprinklers is effective in reducing the negative impact of HS on the physiology and MY of buffalo. This approach is increasingly being used as a replacement for pools, as pool management is much more complicated.

The data from the literature are almost always discordant due to the different sensitivities of the different breeds tested to the different environmental climatic characteristics and different levels of milk production. Most of the data come from studies carried out on buffalo reared under tropical and subtropical conditions. More studies to better understand physiological and metabolic responses to HS and the best mitigation strategies to be adopted for improving welfare and productivity are needed in buffalo breeds reared under temperate climate.

## Figures and Tables

**Figure 1 animals-13-01260-f001:**
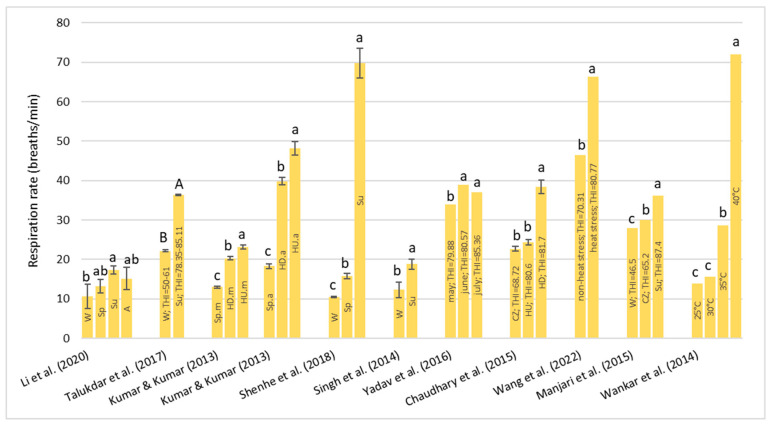
Values of respiration rate (Mean ± SD) reported in the cited literature by several authors. ^a,b,c^
*p* < 0.05; ^A,B^
*p* < 0.01. W = winter; Sp = spring; Su = summer; A = autumn; Sp.m = spring morning; HD.m = hot–dry morning; HU.m = hot–humid morning; Sp.a = spring afternoon; HD.a = hot–dry afternoon; HU.a = hot–humid afternoon; CZ = comfort zone. Li et al. (2000)]: W_THI= 60.25 ± 4.73, Sp_THI = 74.49 ± 4.38, Su_THI = 82.00 ± 0.53, A_THI = 75.51 ± 6.86. Kumar and Kumar (2013): Sp.m_THI = 61.96 ± 0.29, HD.m_THI = 69.21 ± 0.54, HU.m_THI = 78.76 ± 0.12, Sp.a_THI = 70.84 ± 0.25, HD.a_THI = 82.57 ± 0.36, HU.a_THI = 83.68 ± 0.35. Shenhe et al. (2018): W_THI = 49.10 ± 0.50, Sp_THI = 68.11 ± 0.20, Su_THI = 82.95 ± 0.77. Li et al. (2000) [39]; Talukdar et al. (2017) [40]; Kumar & Kumar (2013) [41]; Shenhe et al. (2018) [42]; Singh et al. (2014) [43]; Yadav et al. (2016) [27]; Chaudhary et al. (2015) [45]; Wang et al. (2022) [46]; Manjari et al. (2015) [47]: Wankar et al. (2014) [48].

**Figure 2 animals-13-01260-f002:**
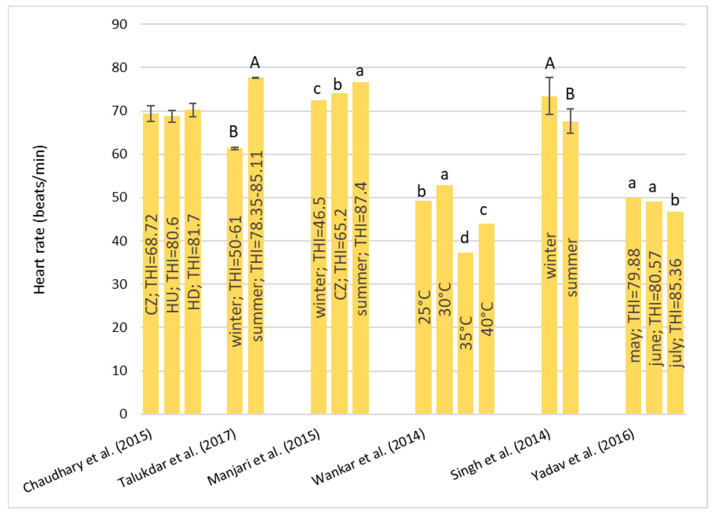
Values of heart rate (mean ± SD) reported in the cited literature by several authors. ^a,b,c,d^
*p* < 0.05; ^A,B^
*p* < 0.01. CZ = comfort zone, HU = hot–humid; HD = hot–dry. Chaudhary et al. (2015) [45]; Talukdar et al. (2017) [40]; Manjari et al. (2015) [47]; Wankar et al. (2014) [48]; Singh et al. (2014) [43]; Yadav et al. (2016) [27].

**Figure 3 animals-13-01260-f003:**
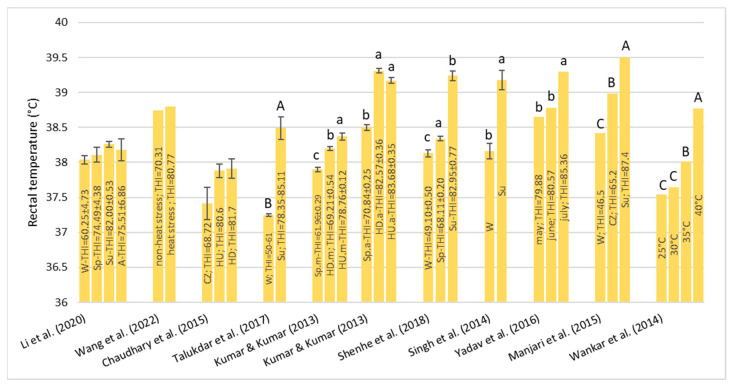
Values of rectal temperature (Mean ± SD) reported in the cited literature by several authors. ^a,b,c^
*p* < 0.05; ^A,B,C^
*p* < 0.01. W = winter; Sp = spring; Su = summer; A = autumn; Sp.m = spring morning; HD.m = hot–dry morning; HU.m = hot–humid morning; Sp.a = spring afternoon; HD.a = hot–dry afternoon; HU.a = hot–humid afternoon; CZ = comfort zone. Li et al. (2000) [39]; Wang et al. (2022) [46]; Chaudhary et al. (2015) [45]; Talukdar et al. (2017) [40]; Kumar & Kumar (2013) [41]; Shenhe et al. (2018) [42]; Singh et al. (2014) [43]; Yadav et al. (2016) [27]; Manjari et al. (2015) [47]: Wankar et al. (2014) [48].

**Figure 4 animals-13-01260-f004:**
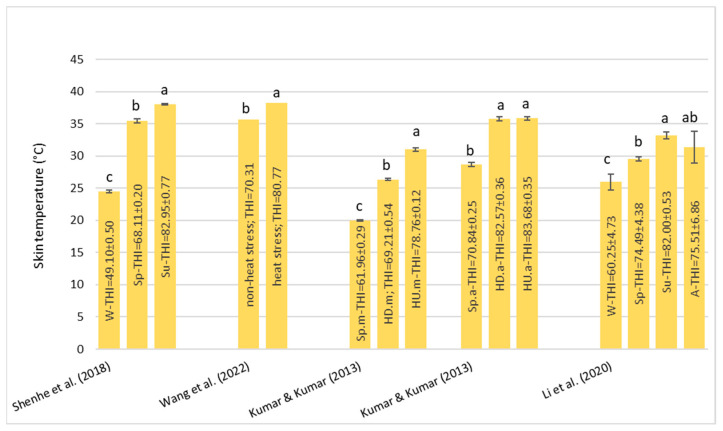
Values of skin temperature (mean ± SD) reported in the cited literature by several authors. ^a,b,c^
*p* < 0.05. W = winter; Sp = spring; Su = summer; A = autumn; Sp.m = spring morning; HD.m = hot–dry morning; HU.m = hot–humid morning; Sp.a = spring afternoon; HD.a = hot–dry afternoon; HU.a = hot–humid afternoon. Shenhe et al. (2018) [42]; Wang et al. (2022) [46]; Kumar & Kumar (2013) [41]; Li et al. (2000) [39].

**Figure 5 animals-13-01260-f005:**
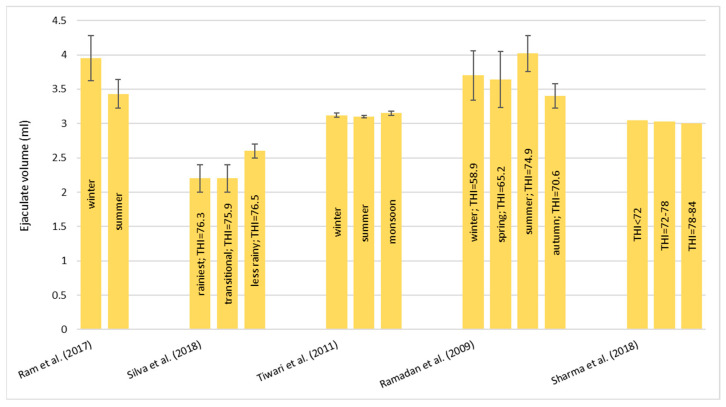
Values of ejaculated volume (Mean ± SD) reported in the cited literature by several authors. Ram et al. (2017) [54]; Silva et al. (2018) [55]; Tiwari et al. (2011) [56]; Ramadan et al. (2009) [57]; Sharma et al. (2018) [58].

**Figure 8 animals-13-01260-f008:**
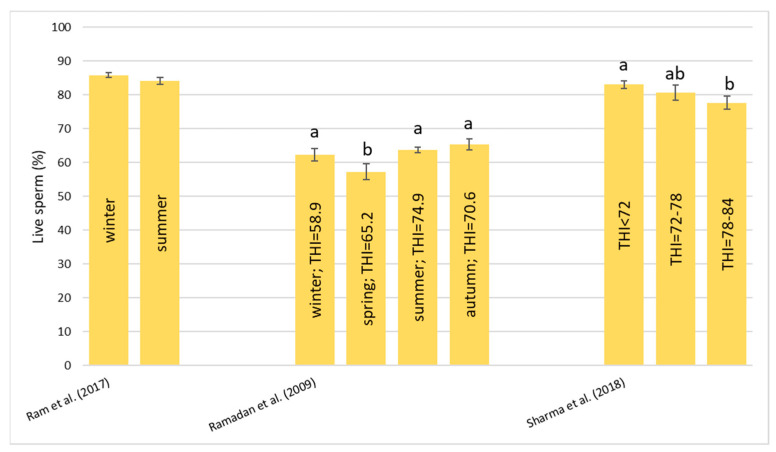
Values of live sperm cells (Mean ± SD) reported in the cited literature by several authors. ^a,b^
*p* < 0.05. Ram et al. (2017) [54]; Ramadan et al. (2009) [57]; Sharma et al. (2018) [60].

**Figure 9 animals-13-01260-f009:**
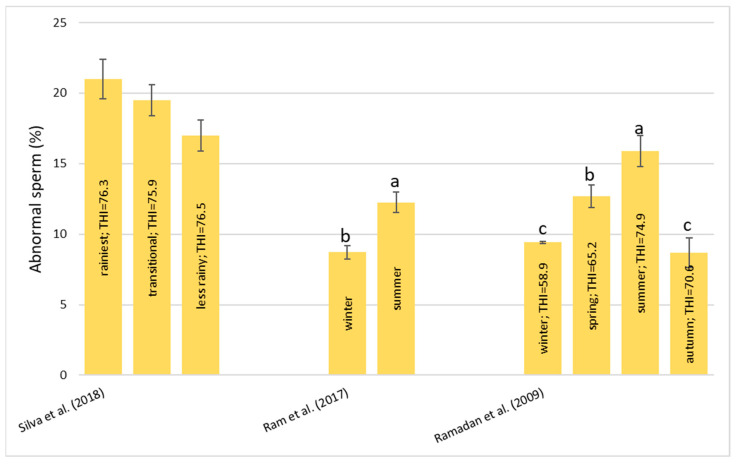
Values of abnormal sperm (Mean ± SD) reported in the cited literature by several authors. ^a,b,c^
*p* < 0.05. Sila et al. (2018) [55]; Ram et al. (2017) [54]; Ramadan et al. (2009) [57].

**Figure 10 animals-13-01260-f010:**
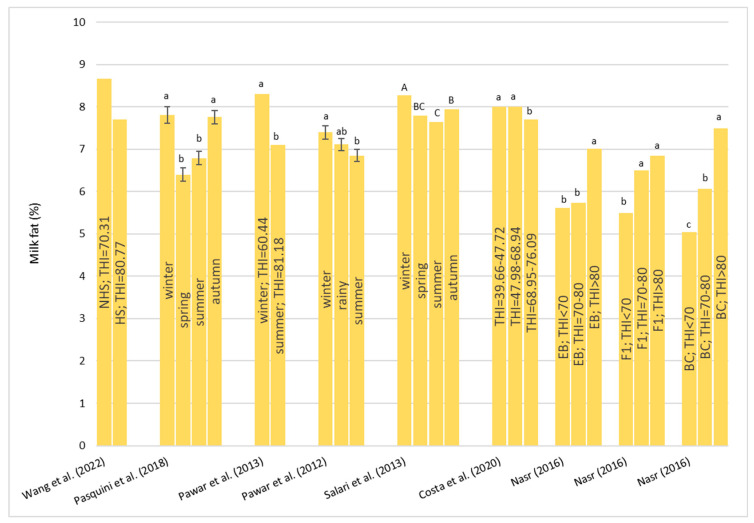
Values of milk fat content (Mean ± SD) reported in the cited literature by several authors. ^a,b,c^
*p* < 0.05, ^A,B,C^
*p* < 0.01. NHS = non-heat-stress; HS = heat stress: EB = pure Egyptian buffalo; F1 = crossbreed Egyptian buffalo 50% × Mediterranean Italian buffalo 50%; BC = backcross of Egyptian buffalo 75% × Mediterranean Italian buffalo 25%. Wang et al. (2022) [46]; Pasquini et al. (2018) [13]; Pawar et al. (2013) [86]; Pawar et al. (2012) [85]; Salari et al. (2013) [84]; Costa et al. (2020) [89]; Nasr (2016) [87].

**Figure 11 animals-13-01260-f011:**
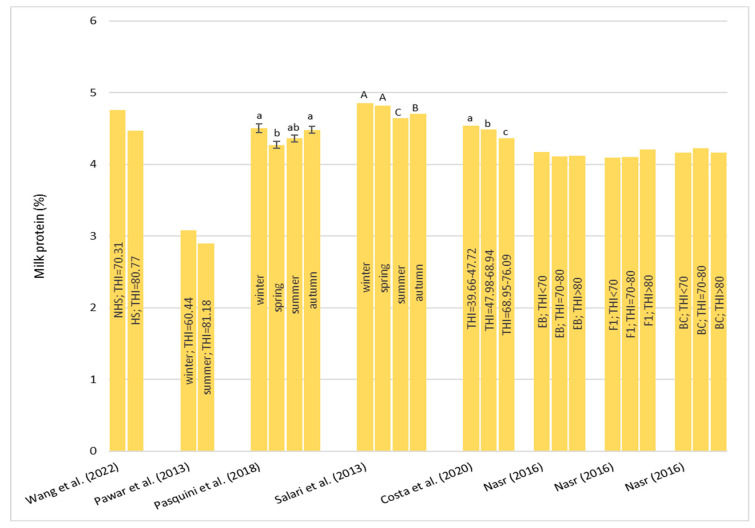
Values of milk protein content (Mean ± SD) reported in the cited literature by several authors. ^a,b,c^
*p* < 0.05, ^A,B,C^
*p* < 0.01. NHS = non-heat-stress; HS = heat stress: EB = pure Egyptian buffalo; F1 = crossbred Egyptian buffalo 50% × Mediterranean Italian buffalo 50%; BC = backcross of Egyptian buffalo 75% × Mediterranean Italian buffalo 25%. Wang et al. (2022) [46]; Pawar et al. (2013) [86]; Pasquini et al. (2018) [13]; Salari et al. (2013) [84]; Costa et al. (2020) [89]; Nasr (2016) [87].

**Figure 12 animals-13-01260-f012:**
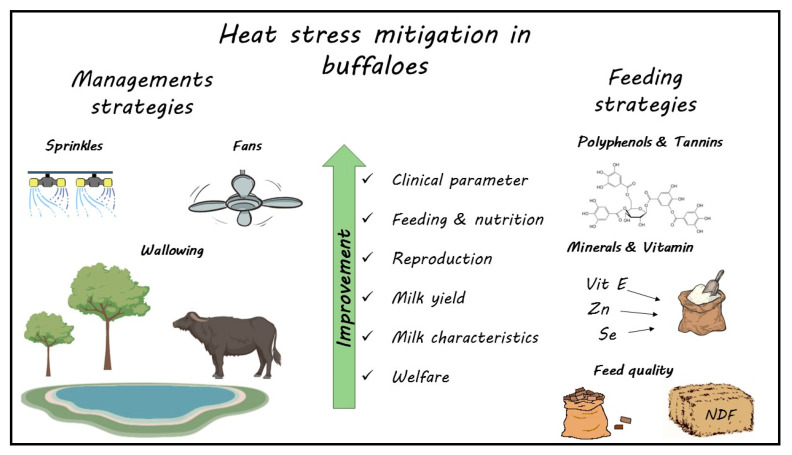
Heat stress mitigation strategies in buffalo.

**Table 1 animals-13-01260-t001:** Physiological response to heat stress in buffalo.

Physiological Response	Effect of Heat Stress	References	Breed
Respiration rate (breaths/min)	↑*	[39]	Nili-Ravi
	↑*	[40]	Murrah
	↑*	[41]	Murrah
	↑*	[42]	
	↑*	[43]	Murrah
	↑*	[44]	Murrah
	↑*	[27]	Murrah
	↑**	[45]	Surti
	↑**	[46]	
	↑**	[47]	Tarai
	↑***	[48]	
Heart rate (beats/min)	n.e.	[45]	Surti
	↑*	[40]	Murrah
	↑**	[47]	Tarai
	↑/↓***	[48]	
	↓*	[43]	Murrah
	↓*	[27]	Murrah
Rectal temperature (°C)	n.e.	[39]	Nili-Ravi
	n.e.	[46]	
	n.e.	[45]	Surti
	↑*	[40]	Murrah
	↑*	[41]	Murrah
	↑*	[42]	
	↑*	[43]	Murrah
	↑*	[44]	Murrah
	↑*	[27]	Murrah
	↑**	[47]	Tarai
	↑***	[48]	
Skin temperature (°C)	↑*	[42]	
	↑*	[46]	
	↑*	[41]	Murrah
	↑*	[39]	Nili-Ravi

n.e. = no effect; ↑ = increase; ↓ = decrease; * *p* < 0.05; ** *p* < 0.01; *** *p* < 0.001.

**Table 3 animals-13-01260-t003:** Effects of heat stress on the semen characteristics of buffalo bulls.

Semen Parameters	Effect of Heat Stress	References	Breed
Ejaculate volume (mL)	n.e.	[54]	Murrah
	n.e.	[55]	Murrah
	n.e.	[56]	Murrah
	n.e.	[57]	Egyptian
	n.e.	[58]	
Sperm concentration (Million/mL)	n.e.	[54]	Murrah
	n.e.	[55]	Murrah
	n.e.	[59]	Murrah
	↓*	[60]	Murrah
	↓**	[57]	Egyptian
Total number of sperm per ejaculate	n.e.	[54]	Murrah
	n.e.	[56]	Murrah
Mass motility (0–5 scale)	↓*	[54]	Murrah
	↓*	[58]	
Initial progressive motility (%)	n.e.	[54]	Murrah
	n.e.	[55]	Murrah
	n.e.	[59]	Murrah
	↑*	[57]	Egyptian
	↓*	[58]	
	↓*	[56]	Murrah
Live sperm (%)	n.e.	[54]	Murrah
	n.e.	[59]	Murrah
	n.e.	[57]	Egyptian
	↓*	[60]	Murrah
Abnormal sperm (%)	n.e.	[55]	Murrah
	↑**	[54]	Murrah
	↑*	[59]	Murrah
	↑**	[57]	Egyptian
Intact acrosome (%)	n.e.	[54]	Murrah
	↓*	[59]	Murrah
HOST (%)	↓**	[54]	Murrah
	n.e.	[60]	Murrah

n.e. = no effect; ↑ = increase; ↓ = decrease; * *p* < 0.05; ** *p* < 0.01; HOST: hypo-osmotic swelling test.

**Table 4 animals-13-01260-t004:** Effects of heat stress on female reproductive parameters.

Parameters	Effect of Heat Stress	References	Breeds
Estrus cycle length (days)	↓	[67]	
Estrus expression (%)	↓	[31]	Murrah
Pregnancy rate—PR (%)	↓	[68]	Mediterranean Italian
	↓***	[65]	Murrah
	↓	[69]	Egyptian
	↓***	[70]	Murrah
Quality of oocytes (%)	↓*	[71]	Egyptian
	↓*	[72]	
	n.e.	[73]	Mediterranean Italian
Days open—DO (days)	↑	[74]	Egyptian (PE)
	↑	[74]	Crossbreed † (F1)
	n.e.	[74]	Backcross# (BC)
	↓*	[75]	Egyptian
Lactation length—LL (days)	n.e.	[75]	Egyptian
	n.e.	[76]	Murrah
	n.e.	[74]	Egyptian (PE)
	↑	[74]	Crossbreed † (F1)
	↑	[74]	Backcross# (BC)
	↓	[77]	Murrah
	↓**	[78]	Anatolian
	↓**	[79]	Anatolian
Calving interval—CI (days)	n.e.	[74]	Egyptian (PE)
	↑	[74]	Crossbreed † (F1)
	↑	[74]	Backcross# (BC)
	n.e.	[77]	Murrah
	n.e.	[78]	Anatolian
	↓*	[75]	Egyptian
	↓*	[79]	Anatolian
	↓*	[76]	Murrah
Dry period—DP (days)	n.e.	[77]	Murrah
	↓*	[75]	Egyptian
	↓*	[76]	Murrah
Conception rate—CR (%)	↓***	[70]	Murrah
	↓	[31]	Murrah
	↓	[74]	Egyptian (PE)
	↓	[74]	Crossbreed (F1)
	↓	[74]	Backcross (BC)
Service period—SP (days)	n.e.	[76]	Murrah
	↑	[70]	Murrah
	↑	[35]	Murrah

n.e. = no effect; ↑ = increase; ↓ = decrease; * *p* < 0.05, ** *p* < 0.01; *** *p* < 0.001; PE: pure Egyptian; F1 crossbreed Egyptian buffalo 50% × Mediterranean Italian buffalo 50%; BC backcross of Egyptian buffalo 75% × Mediterranean Italian buffalo 25%.

**Table 5 animals-13-01260-t005:** Effects of heat stress on milk yield.

Parameter	Effect of Heat Stress	References	Breeds
Milk Yield (L)	↓*	[88]	Mediterranean Italian
	n.e.	[84]	Mediterranean Italian
	n.e.	[68]	Mediterranean Italian
	n.e.	[23]	Mediterranean Italian
	↓	[31]	Murrah
	↓*	[85]	Murrah
	↓*	[86]	Murrah
	↓	[34]	Murrah
	n.e.	[87]	Egyptian (PE)
	↓*	[87]	Crossbreed (F1)
	n.e.	[87]	Backcross (BC)

n.e. = no effect; ↑ = increase; ↓ = decrease; F1 Egyptian buffalo 50% × Mediterranean Italian buffalo 50%, BC Egyptian buffalo 75% × Mediterranean Italian buffalo 25%, * *p* < 0.05; ** *p* < 0.01; *** *p* < 0.001.

**Table 6 animals-13-01260-t006:** Effects of heat stress on milk characteristics and composition.

Milk Parameters	Effect of Heat Stress	References	Breed
Fat (%)	n.e.	[46]	
	↓*	[13]	Mediterranean Italian
	↓*	[86]	Murrah
	↓*	[85]	
	↓**	[84]	Mediterranean Italian
	↓***	[23]	Mediterranean Italian
	↓***	[89]	Mediterranean Italian
	↑*	[87]	
Protein (%)	n.e.	[86]	Murrah
	n.e.	[46]	
	n.e.	[87]	
	n.e.	[13]	Mediterranean Italian
	↓**	[84]	Mediterranean Italian
	↓***	[89]	Mediterranean Italian
	↑**	[23]	Mediterranean Italian
Lactose (%)	n.e.	[87]	
	↓***	[89]	Mediterranean Italian
	↑*	[13]	Mediterranean Italian
Milk urea nitrogen (mg/dL)	n.e.	[46]	
	↑***	[89]	Mediterranean Italian
Elecrical conductivity (mS)	↓**	[89]	Mediterranean Italian
pH	↓***	[89]	Mediterranean Italian
Somatic cells count (n/mL)	n.e.	[89]	Mediterranean Italian
	n.e.	[46]	
	↑***	[23]	Mediterranean Italian
	n.e.	[84]	Mediterranean Italian
	n.e.	[13]	Mediterranean Italian
Total bacteria count (n/mL)	↑*	[89]	Mediterranean Italian
	n.e.	[13]	Mediterranean Italian
Rennet coagulation time (min)	n.e.	[89]	Mediterranean Italian
Curd firmness (mm)	↑*	[89]	Mediterranean Italian
Curd firming time (min)	n.e	[89]	Mediterranean Italian

n.e. no effect; ↑ increase; ↓ decrease; * *p* < 0.05; ** *p* < 0.01; *** *p* < 0.001.

**Table 7 animals-13-01260-t007:** Effect of different nutritional and feeding treatment on several parameters.

References	Breed	Treatment	RT	ST	RR	HR	DMI	MY	MQ
[90]	Med. Italian	Zn + Se					n.e.	↑*	n.e.
[91]	Surti	Vit B_3_	↓*	↓*	↓*		n.e.	↑*	↑*
[93]	Egyptian	*Moringa oleifera* leaves	↓*	↓*	↓*	↓*			
[94]	Murrah	34.5% NDF and 7% MP	n.e.		↓*	n.e.	n.e.	n.e.	↑*
		37% NDF and 7% MP	n.e.		n.e.	↓*	n.e.	n.e.	↑*
		30% NDF and 8% MP	↓*		↓*	↓*	n.e.	n.e.	↑*
		34.5% NDF and 8% MP	↓*		↓*	↓*	n.e.	n.e.	↑*
		37% NDF and 8% MP	↓*		↓*	↓*	n.e.	n.e.	↑*
[40]	Murrah	+15% ME	n.e.		n.e.	n.e.			
		−15% ME	↑*		↑*	n.e.			

n.e. no effect; ↑ increase; ↓ decrease; * *p* < 0.05. RT rectal temperature, ST skin temperature, RR respiration rate, HR heart rate, DMI dry matter intake, MY milk yield, MQ milk quality, NDF neutral detergent fiber, MP metabolizable protein, and ME metabolizable energy.

**Table 8 animals-13-01260-t008:** The effect of different cooling strategies on several parameters.

References	Breed	Treatment Type	FB	CP	SI	MY	MC	RC	DMI	WI
[88]	Mediterranean Italian	Free stall open-sided barn with an outdoor lot and a concrete pool	n.e.		↑**	↑*	n.e.	n.e.		
[98]	Nili-Ravi	Shade with a fan				↑*	↑*		↑*	↓*
		Shade, a fan, and sprinklers				↑*	↑*		↑*	↓*
[27]	Murrah	Misting		↓*		↑*				
		Wallowing		↓*		n.e.				
[101]	Murrah	Silvopastoral system	↑*	↓*						
[97]	Murrah	Wallowing		↓*		↑*			↑*	
[99]	Nili-Ravi	Shade with a fan		↓*		n.e.	↑*		↑*	↓*
		Shade, a fan, and sprinklers		↓*		↑*	↑*		↑*	↓*

n.e. = no effect; ↑ = increase; ↓ = decrease; * *p* < 0.05; ** *p* < 0.01. FB feeding behavior, CP clinical parameters, MY milk yield, SI social interaction, RC reproductive characteristics, SQ semen quality, DMI dry matter intake, and WI water intake.

## Data Availability

Data sharing not applicable.

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
