# Peer review of "Responses of Dairy Buffalo to Heat Stress Conditions and Mitigation Strategies: A Review"

_animals, 2023, doi:10.3390/ani13071260_

Round 1

Reviewer 1 Report

Dear authors.

The manuscript you have submitted for review deals with an interesting topic and is developed meticulously, covering different physiological and productive variables. In general, the manuscript is well written, and the authors make a detailed review of the subject. The manuscript is extensive, with a large amount of data that the authors present in their work. This length leads the reader to lose the direction proposed by the authors in the drafting of the manuscript. The authors should close the major sections with a summary of the major findings. This would make the manuscript easier to read.

The dispersion of data that is provided in many of the bibliographic references used by the authors, whether regarding age, race, or any of the individual factors of the works used as a basis, makes the conclusions of this manuscript ambiguous. The authors could specify these cases and discuss them further so that readers can read along with the development of the idea of the review.  

The graphs shown by the authors do not make the reader visualize in a good way the data that the authors want to highlight. I suggest arrange the axes and their titles of them. They should make the captions more explicit in order to understand the graphs. Moreover, in the text of the manuscript, the graphs should be better used, as they are mentioned, but in most cases, they do not complement the description of the subject. The tables are well laid out, and manage to highlight the important data to be shown.

Finally, I find this a very accurate and good review, so I hope that the authors will integrate my comments into the text, to improve the readability of the manuscript. I decide that I approve with major revision.

Author Response

We would like to thank the Reviewer for his/her effort in reviewing the manuscript and useful comments and suggestions 

R_1

The manuscript you have submitted for review deals with an interesting topic and is developed meticulously, covering different physiological and productive variables. In general, the manuscript is well written, and the authors make a detailed review of the subject.

We thank the R for his/her positive comments.

The manuscript is extensive, with a large amount of data that the authors present in their work. This length leads the reader to lose the direction proposed by the authors in the drafting of the manuscript. The authors should close the major sections with a summary of the major findings. This would make the manuscript easier to read.

Thanks for comment. Summaries have been added and all the specification are included for better understanding the manuscript.

The dispersion of data that is provided in many of the bibliographic references used by the authors, whether regarding age, race, or any of the individual factors of the works used as a basis, makes the conclusions of this manuscript ambiguous. The authors could specify these cases and discuss them further so that readers can read along with the development of the idea of the review. 

The graphs shown by the authors do not make the reader visualize in a good way the data that the authors want to highlight. I suggest arrange the axes and their titles of them. They should make the captions more explicit in order to understand the graphs. Moreover, in the text of the manuscript, the graphs should be better used, as they are mentioned, but in most cases, they do not complement the description of the subject.

Thanks for the suggestion. Titles of axis have been arranged and the figures have been better integrated into the text.

The tables are well laid out, and manage to highlight the important data to be shown.

Many thanks

Finally, I find this a very accurate and good review, so I hope that the authors will integrate my comments into the text, to improve the readability of the manuscript. I decide that I approve with major revision.

Many thanks. We tried to respond and integrate the manuscript with indication and comment suggested.

Reviewer 2 Report

The review covers all the important topics of heat stress on both sexes.

The research mainly addresses the impact of heat stress on reproductive variables of Buffaloes.  The information about buffaloes management is limited. This review will help to understand the Buffalo dairy industry.  It is a review and addresses important information about weather climate change and how it might impact animal production (Buffaloes). The references are appropriate.

Author Response

R_2

The review covers all the important topics of heat stress on both sexes.

The research mainly addresses the impact of heat stress on reproductive variables of Buffaloes.  The information about buffaloes management is limited. This review will help to understand the Buffalo dairy industry.  It is a review and addresses important information about weather climate change and how it might impact animal production (Buffaloes). The references are appropriate.

We thank the R for his/her comments.

Reviewer 3 Report

Rewiew Animals-2285592

In this review, the authors extensively discuss the repercussions of Heat Stress (HS) on the main buffalo productions (milk quality, reproduction and animal health). Indeed; it is well known that rising temperatures and climate change worldwide have been ongoing for a long time and many countries have implemented strategies to counteract these effects in order to improve animal welfare and production. The authors focus on possible mitigation systems such as genetic improvement (thermotolerance), nutritional strategies (vitamins, minerals, antioxidants, etc.) and structural improvements on livestock farms to cool the environment in which the animals are reared.

This review, in line with the subject matter of the journal, is well articulated in its treatment of the topics, with great interest for the scientific community, it needs some minor additions that I report below line-by-line: 

Line 48: add reference year. 

Line 56: I suggest specifying the differences between 'Mozzarella di Bufala' and 'Mozzarella di Bufala Campana PDO'. 

Line 76- 77: “The milk production and its chemical composition are affected by several factors like breed, genetics, number of lactations, days of lactation, lactation phase, feeding, season, etc. [12,13].”. I suggest that the authors include also “health status”. 

Line 678 and 688: rephrase as:  Mediterranean Italian buffalo.

Line 732: The paper "Costa et. al, 2021; ref n. 89" is structured on the analysis of "bulk milk" samples (obtained from at least 2 milkings), it should be specified in the text. 

t is essential to consider that the coagulation characteristics of milk differ between individual milk (1 milking from a single animal) and mass milk (at least 2 milkings from several animals), for different reasons, including the total bacterial count levels

Line 741-747: I suggest the authors integrate the discussion made on "Milking Clotting Properties", it is essential to consider that the coagulation characteristics of milk differ between individual milk (1 milking from each animal) and bulk milk (at least 2 milkings from several animals), for different reasons, including the Total Bacterial Count levels.

I suggest the authors double-check the graphs, especially the letters indicating significant differences, e.g. graph 1 'Lie et al (39)', in the first and third histograms perhaps 'ab' should be inserted.

References: n. 7. AIA Bulletin, 2021. I suggest indicating 'website' access.

Author Response

We would like to thank the R for his/her effort in reviewing the manuscript and useful comments and suggestions.

R_3

Comments

Solutions

Line 48: add reference year. 

Thanks for the suggestion. Year of the reference has been added.

Line 56: I suggest specifying the differences between 'Mozzarella di Bufala' and 'Mozzarella di Bufala Campana PDO'

Thanks for the comment. This part has been improved.

Line 76- 77: “The milk production and its chemical composition are affected by several factors like breed, genetics, number of lactations, days of lactation, lactation phase, feeding, season, etc. [12,13].”. I suggest that the authors include also “health status”.

Thanks for the suggestion. “Health status” has been added.

Line 678 and 688: rephrase as:  Mediterranean Italian buffalo.

Thanks for specification. The name has been corrected throughout the manuscript.

Line 732: The paper "Costa et. al, 2021; ref n. 89" is structured on the analysis of "bulk milk" samples (obtained from at least 2 milkings), it should be specified in the text. 

Thanks for the comment. This part has been improved as suggested.

It is essential to consider that the coagulation characteristics of milk differ between individual milk (1 milking from a single animal) and mass milk (at least 2 milkings from several animals), for different reasons, including the total bacterial count levels

Thanks for specification. The discussion has been integrated with the reviewer suggestions and comments.

Line 741-747: I suggest the authors integrate the discussion made on "Milking Clotting Properties", it is essential to consider that the coagulation characteristics of milk differ between individual milk (1 milking from each animal) and bulk milk (at least 2 milkings from several animals), for different reasons, including the Total Bacterial Count levels.

I suggest the authors double-check the graphs, especially the letters indicating significant differences, e.g. graph 1 'Lie et al (39)', in the first and third histograms perhaps 'ab' should be inserted.

Thanks for the comment. Graphs have been corrected.

References: n. 7. AIA Bulletin, 2021. I suggest indicating 'website' access.

Thanks for the suggestion. Website access of the reference has been added.

Round 2

Reviewer 1 Report

Dear authors.

Thank you for including my comments and observations in your manuscripts.